# The Dynamics of *Cryptococcus neoformans* Cell and Transcriptional Remodeling during Infection

**DOI:** 10.3390/cells11233896

**Published:** 2022-12-02

**Authors:** Gustavo J. C. Freitas, Ludmila Gouveia-Eufrasio, Eluzia C. P. Emidio, Hellem C. S. Carneiro, Ludmila de Matos Baltazar, Marliete C. Costa, Susana Frases, Glauber R. de Sousa Araújo, Tatiane A. Paixão, Brunno G. Sossai, Melissa Caza, James W. Kronstad, Nalu T. A. Peres, Daniel A. Santos

**Affiliations:** 1Laboratório de Micologia, Departamento de Microbiologia, Universidade Federal de Minas Gerais, Belo Horizonte 31270-901, Brazil; 2Instituto de Patologia Tropical e Saúde Pública, Universidade Federal de Goiás, Goiânia 74605-020, Brazil; 3Laboratório de Ultraestrutura Celular Hertha Meyer, Instituto de Biofísica Carlos Chagas Filho, Universidade Federal do Rio de Janeiro, Rio de Janeiro 21941-902, Brazil; 4Laboratório de Patologia Celular e Molecular, Departamento de Patologia, Universidade Federal de Minas Gerais, Belo Horizonte 31270-901, Brazil; 5Michael Smith Labs, University of British Columbia, Vancouver, V6T 1Z4, Canada

**Keywords:** cell size, ribosome biogenesis, inositol pathway, proteasome, cell remodeling, cryptococcosis

## Abstract

The phenotypic plasticity of *Cryptococcus neoformans* is widely studied and demonstrated in vitro, but its influence on pathogenicity remains unclear. In this study, we investigated the dynamics of cryptococcal cell and transcriptional remodeling during pulmonary infection in a murine model. We showed that in *Cryptococcus neoformans*, cell size reduction (cell body ≤ 3 µm) is important for initial adaptation during infection. This change was associated with reproductive fitness and tissue invasion. Subsequently, the fungus develops mechanisms aimed at resistance to the host’s immune response, which is determinant for virulence. We investigated the transcriptional changes involved in this cellular remodeling and found an upregulation of transcripts related to ribosome biogenesis at the beginning (6 h) of infection and a later (10 days) upregulation of transcripts involved in the inositol pathway, energy production, and the proteasome. Consistent with a role for the proteasome, we found that its inhibition delayed cell remodeling during infection with the H99 strain. Altogether, these results further our understanding of the infection biology of *C. neoformans* and provide perspectives to support therapeutic and diagnostic targets for cryptococcosis.

## 1. Introduction

*Cryptococcus neoformans* is the leading agent of cryptococcosis, an invasive fungal infection that occurs by inhaling desiccated yeasts or basidiospores dispersed in the environment [1,2]. The yeast initially settles in the lung, causing pneumonia, and then spreads to other organs, such as the central nervous system (CNS), leading to cryptococcal meningitis, the most severe form of the disease [2]. Cryptococcosis is responsible for 152,000 new cases of meningitis annually, resulting in 112,000 deaths [3].

The cells of *Cryptococcus* spp. switch morphology in response to environmental conditions, a plasticity that may enhance survival. Several morphotypes have been described, such as hyphae, pseudohyphae, seed cells, microcells (<2 µm), typical yeast cells (6–8 µm), and titan cells (>10 µm) [4,5,6,7]. Desiccated yeast cells or spores are thought to be the morphological types that are inhaled to initiate infection [8]. Cells of *Cryptococcus* spp. are generally found in the yeast phase inside the host, presenting as small cells, typical cells, and titan cells. In this context, titan cells are often associated with higher virulence [9,10,11]. However, titan cells are a minority of cryptococcal cells and are rarely observed outside of the lungs. Considering that most studies are focused on morphological assessment at late stages of infection, little is known about the kinetics of in vivo morphological transitions at the beginning of infection and how this may be determinant for fungal survival and disease progression. The size and morphology can affect the virulence of pathogenic microorganisms. For example, the reduction in bacterial size allows evasion of the immune response by *Streptococcus pneumoniae* [12]. In *Candida albicans*, hyphae are essential for tissue invasion, and yeast cells are crucial for hematogenous and lymphatic dissemination [13,14]. More recently, the seed cells of *C. neoformans* have been shown to be better at spreading into extrapulmonary tissues and in surviving intracellularly [6,15]. However, further studies on the morphophysiological adaptation of *C. neoformans* in the bronchoalveolar space and in the pulmonary epithelium are still needed.

The polysaccharide capsule is considered the main virulence factor of the *Cryptococcus* genus [16]. In the environment, it confers resistance to desiccation and phagocytosis by free-living amoebae, while in the host, it has antioxidant and antiphagocytic roles in addition to the modulating of the host’s immune response [17]. Melanin, another determining factor of virulence, is associated with resistance to UV radiation and oxidative stress and has antiphagocytic properties [18]. The synthesis of the polysaccharide capsule and melanin may vary according to the culture condition, the time of infection, and the anatomical site where the yeast is found [19,20,21,22,23]. Usually, studies evaluate these virulence factors using different in vitro/ex vivo culture conditions or infection models [6,24,25] that do not reproduce the natural route of infection (lung–blood–brain). Thus, it is necessary to establish whether the in vitro/ex vivo phenotypes are observed in vivo, and how they impact fungal virulence and the course of the infection. For instance, the VGI genotype has been associated with a thicker polysaccharide capsule in vitro. However, this is not the only parameter that impacts virulence, since strains of the same genotype may have different virulence profiles [22,26].

In this study, we observed the morphological, physiological, and transcriptional reorganization of *C. neoformans* in the bronchoalveolar space and pulmonary epithelium throughout the infection, which we call cellular and transcriptional remodeling. We demonstrate that this is a dynamic process during infection and promotes different levels of virulence. An analysis of fungal and host transcriptomes during infection revealed that fungal transcriptional response is initially focused on strategies to adapt and reproduce during host colonization. Subsequently, transcriptional patterns reflected cellular mechanisms focusing on resistance to the immune response and fungal virulence. The murine transcriptional response was consistent with phagocytosis and an inflammatory response. We also identified transcriptional profiles in the early and late stages of infection in fungi and mice.

## 2. Materials and Methods

### 2.1. Fungal Strains and Media

Five strains of *C. neoformans* were used, including the reference strain H99 from the Duke University Medical Center, North Carolina (United States), and four strains representing other genotypes/serotypes (WM 626, WM 628, WM 629, and WM 148) (Table 1). All strains were maintained at −80 °C and were cultured on Yeast Extract Peptone Dextrose (YPD—2% glucose, 2% peptone, and 1% yeast extract) or M = minimal medium (MM—15 mM glucose, 10 mM MgSO_4_·7H_2_O, 29.4 mM KH_2_PO_4_, 13 mM glycine, and 3 mM thiamine-HCl, pH 5.5) for 48 h/72 h (exponential phase) at 37 °C for each experiment.

### 2.2. Phenotypic Characterization In Vitro

Growth curves, pigmentation, morphology, laccase, and urease activity were analyzed for each strain. Initially, to validate the growth analysis in the spectrophotometer, we compared the OD600 data with the plating in culture medium for the H99 strain. We observed that the OD600 results reproduced the profile observed when CFU was determined (data not shown). Thus, for experimental optimization, we used the OD600 analysis for the growth curve of the other strains. For this, 5 × 10^5^ cells/mL of each strain were dispensed into a 96-well plate with liquid YPD or MM and incubated for 72 h in a spectrophotometer (OD 600 nm) at 37 °C. For each strain, eight replicates were performed. The area under the curve (AUC) was used to compare the groups. Melanin production was visually determined by growing the strains in solid MM supplemented with 1 mM L-3,4-dihydroxphenylalanine L-DOPA (Sigma-Aldrich, Burlington, MA, USA) and incubation for five days at 37 °C. Laccase and urease activity were quantified as previously described [27,28,29,30].

For the morphometric analysis, the strains were cultivated in YPD and MM broth for 72 h at 37 °C. After incubation, yeasts were suspended on a slide with India ink, followed by visualization under an optical microscope and image capture. Cell body and capsule sizes of at least 100 specimens from each condition were measured using the Image J program (http://rsb.info.nih.gov/ij/ (accessed on 4 September 2020); National Institutes of Health, NIH, Bethesda, MD, USA) (Araujo et al., 2012). Cell body diameter was defined as the diameter without the capsule. Capsule size was calculated by the ratio of capsule thickness to cell body radius. The total cell size was defined as the diameter of the cell body, including the capsule. In addition, the surface/volume ratio was determined using the formula 3/r, where r = radius [21]. All phenotypic assays were performed in triplicate.

### 2.3. Mice Experimentation

#### 2.3.1. Ethics Statement

This work was approved (protocol 235/2017) by the Ethics Committee in the Use of Animals (CEUA) from Universidade Federal de Minas Gerais. We followed the Brazilian Society of Zootechnics/Brazilian College of Animal Experimentation guidelines (available online http://www.cobea.org.br/ (accessed on 10 January 2019)) and Federal Law 11,794. Water and food were provided ad libitum and light/dark cycles were maintained. All efforts to minimize the suffering of the animals were carried out.

#### 2.3.2. Mice Survival and Behavior

The inoculum of 1 × 10^5^ CFU/30 µL of each strain was used to infect C57/BL6 male mice. Inoculum preparation was based on the Neubauer chamber count of viable cells stained with Trypan Blue. Animals were infected intratracheally under anesthesia with ketamine (100 mg/kg) and xylazine (16 mg/kg) [31,32]. Animals were monitored daily for survival analysis and behavior assessment using the SmithKline/Harwell/ImperialCollege/Royal Hospital/PhenotypeAssessment (SHIRPA) protocol. This protocol provides reliable information about murine brain dysfunction and its general status. The individual parameters evaluated were grouped into five functional categories: neuropsychiatric status, motor behavior, autonomic function, muscle tone and strength, and reflex and sensory function. The score for each category was calculated as previously described [32]. Appendix A describes the parameters analyzed.

According to the mean lethality time (MLt), the strains were classified as hypervirulent (MLt = 21–25 days), virulent (MLt = 26–31 days), hypovirulent (MLt = 32–40 days), and non-lethal (MLt > 100 days).

#### 2.3.3. Fungal Burden

After analysis of survival and behavior, another group of mice was anesthetized and infected intratracheally to assess fungal burden in the bronchoalveolar space (BS), the pulmonary epithelium (PE), and the brain after 10 days of infection. Animals were euthanized under anesthesia, and the organs and bronchoalveolar lavage fluid (BALF) were aseptically removed to quantify the fungal burden in the BS. BALF was isolated by inserting a catheter in the trachea of terminally anesthetized mice, through which 1 mL PBS solution was instilled into the bronchioles. The fluid was gently retracted to maximize BAL fluid retrieval. Lungs and brains were weighed and ground in Petri dishes with 1 mL sterile PBS. Then, the suspensions were cultivated on YPD agar for 48 h at 37 °C. Colonies were counted and the results were expressed in CFU/g or CFU/mL.

#### 2.3.4. Macrophage Assays

Bone-marrow-derived macrophages (BMDM) were used to evaluate the susceptibility of *C. neoformans* strains to the fungicidal activity of macrophages. Briefly, bone marrow cells were harvested from the tibias and femurs of C57BL/6 male mice. Then, cells were cultured in BMDM differentiation medium (RPMI (Gibco, Thermo Fisher Scientific, Waltham, MA, USA) supplemented with 30% L929 growth-conditioning media, 20% fetal bovine serum (Gibco), 2 mM glutamine (Sigma-Aldrich, Burlington, MA, USA), 100 units/mL of penicillin-streptomycin (Gibco, Thermo Fisher Scientific, Waltham, MA, USA), 50 µM of 2-mercaptoethanol (Gibco, Thermo Fisher Scientific, Waltham, MA, USA)) for 1 week at 37 °C/5% CO_2_ (25). Aderant cells differentiated into macrophages were resuspended and transferred to a sterile polypropylene tube. BMDMs were centrifuged at 200× *g*/5 min at 4 °C and resuspended in 5 mL of RPMI 1640 medium containing 10% BFS, 2 mM glutamine, 25 mM HEPES (Gibco, Thermo Fisher Scientific, Waltham, MA, USA) pH 7.2, 100 units/mL G penicillin, and 5% L929 cell culture supernatant. Cell viability was determined with Trypan Blue (Sigma-Aldrich, Burlington, MA, USA), followed by plating in 24-well plates for phagocytosis percentage (PP) and fungicidal activity (FA) determination, or in 96-well plates for reactive oxygen species (ROS) and peroxynitrite quantification (PRN) [32].

PP and killing assays were determined after 2 and 24 h of BMDM infection. Briefly, for PP analyses, glass slides were placed at the bottom of 24-well plates, and cells were plated and infected in a proportion of 5:1 (yeast:macrophages). After 2 and 24 h of infection, the glass coverslips were removed and stained with Panotico Rapido dye (Laborclin, Pinhais, Parana, Brazil). PP was determined by counting the percentage of BMDM with internalized yeasts under an optical microscope. For the killing assay, supernatants were removed, and non-internalized and non-adherent yeast cells were removed by two washes with PBS. BMDMs were lysed with 200 μL of sterile distilled water for 30 min at 37 °C [32]. Both phagocytosis and killing assays were performed in six replicate techniques, and the data presented represent three independent experiments.

To quantify ROS and PRN, 2,7-dichlorofluorescein diacetate (DCFH-DA; Invitrogen, Thermo Fisher Scientific, Waltham, MA, USA) and dihydrorhodamine 123 (DHR 123; Invitrogen, Thermo Fisher Scientific, Waltham, MA, USA) were used, respectively. After incubation of the dyes with infected BMDMs for 3 and 24 h at 37 °C, the fluorescence was read at excitation wavelengths of 485 nm and emission of 530 nm. Data were expressed as arbitrary fluorescence units (AU) ± SE [32,33,34] representing the levels of intracellular ROS and PRN of the fungus and macrophage.

#### 2.3.5. Kinetics of *C. neoformans* Morphology In Vivo

After analysis of survival, one strain representing each virulence profile (H99—hypervirulent, WM628—virulent, WM148—hypovirulent, and WM626—non-lethal) was selected for the morphological evaluation of fungal cells at different times of infection (0, 6 h, 24 h, and 240 h). Time 0 represents the yeasts used for infection, previously cultured in YPD. Animals were infected intratracheally and euthanized at the established times. BALF, lungs, and brain were removed, ground, and fixed on a slide with India ink, followed by visualization under an optical microscope and image capture. Cell and capsule sizes of at least 100 yeasts from each condition were measured using the Image J program (http://rsb.info.nih.gov/ij/ (accessed on 4 January 2021); National Institutes of Health, NIH, Bethesda, MD, USA) [17]. In addition, BALF was also used for scanning electron microscopy and RNA analyses.

#### 2.3.6. Scanning Electron Microscopy of BALF

BALF was fixed in 2.5% glutaraldehyde type I, in 0.1 M sodium cacodylate buffer (pH 7.2), for 1 hour at room temperature. After fixation, cells were washed in 0.1 M sodium cacodylate buffer (pH 7.2) containing 0.2 M sucrose and 2 mM MgCl_2_. Cell surface was observed by scanning electron microscopy (SEM) [35]. Obtained images were colored using Photoshop software.

#### 2.3.7. Kinetics of Fungal Burden in the Brain after Intratracheal Infection

Analysis of fungal burden in the brain after 6 h, 24 h, and 240 h was performed. Animals were infected intratracheally and euthanized at the established times. The tissue was weighed and ground in Petri dishes with 1 mL of sterile PBS. Then, the solution was cultured in YPD agar and incubated for 48 h at 37 °C. Colonies were counted, and results expressed as CFU/g.

#### 2.3.8. Histopathology of Brain

To evaluate the melanin phenotype in the brain, mice were infected by intracranial inoculation. An inoculum of 1 × 10^2^ CFU/5 µL was prepared and counted in a Neubauer chamber with Trypan Blue. Only the strains that did not produce melanin in vitro (WM628 and WM626) were analyzed in this assay, and H99 was used as control. After preparing the inoculum, the animals were anesthetized and immobilized, and intracranial infection was performed with a 30-gauge needle, fixed to an insulin syringe with a cuff to prevent penetration > 1 mm. A midline puncture through the cranial vault was made 6 mm posterior to the orbit, and the inoculum was injected [36]. After 10 days of infection, the animals were euthanized, and their brains removed. The tissue was fixed in formalin, embedded in paraffin, and then subjected to Fontana–Masson (FMS) staining [37]. After microscopic analysis, the presence of melanin was considered as the brown to black color in the yeast.

#### 2.3.9. Dual RNA Sequencing

##### Sample Preparation

C57BL/6 mice were infected intratracheally with the H99 strain previously grown in YPD to analyze the transcriptome. BALF obtained from 6 mice of each condition was pooled together, generating two replicates at 6 h and 240 h (10 d). BALF was centrifuged at 1200× *g* and resuspended in 1 mL TRIzol Reagent (Invitrogen, Thermo Fisher Scientific, Waltham, MA, USA) for RNA isolation, according to the manufacturer’s protocol. RNA from H99 yeasts grown in YPD was used as control.

##### Library Preparation and Sequencing

RNA quality, library preparation, sequencing reactions, and initial bioinformatics analyses were conducted at GENEWIZ Inc. (South Plainfield, NJ, USA). RNA samples were quantified by fluorometry (Qubit 2.0, Thermo Fisher Scientific, Waltham, MA, USA), and RNA integrity was checked with a 2100 Bioanalyzer (Agilent Technologies, Santa Clara, CA, USA). Transcriptome sequencing was performed via rRNA depletion using the Illumina HiSeq platform (Illumina, San Diego, CA, USA) in the 2 × 150 bp Paired-End configuration.

##### Data Analysis

Quality reads were filtered using the FastQC software, and adapters were removed by the Trimmomatic v.0.36 [38]. Trimmed reads were mapped to the *C. neoformans* H99 or the Mus musculus GRCm38 reference genomes, both available on ENSEMBL, using the STAR aligner v.2.5.2b [39]. STAR aligner detects splice junctions, incorporating them to align the entire read sequences and generating the BAM files. Table 3 show the statistics of genome mapping. Unique gene hit counts were calculated using feature Counts from the Subread package v.1.5.2 [40], and unique reads that fell within exon regions were counted. Differentially expressed genes (DEGs) were obtained using a 5% false discovery rate (FDR), using the DESeq2 Bioconducter package [41].

##### GO Analysis and Regulatory Network Constructions

Fungal DEGs were functionally categorized using Gene Ontology (GO) and KEGG (Kyoto Encyclopedia of Genes and Genomes), using FungiDB ((http://fungidb.org/fungidb/ (accessed on 5 March 2020)) and FungiFun2 (https://elbe.hki-jena.de/fungifun/ (accessed on 5 March 2020)). Mouse DEGs were categorized using the ShinyGO v0.75 database [42]. Highly represented categories under each condition were determined by enrichment analysis. All data were submitted in the GEO (Gene Expression Omnibus) database GSE206758.

#### 2.3.10. *C. neoformans* (H99) Morphology in Mice under Treatment with a Proteasome Inhibitor

After analyzing the transcriptome results, a new infection was performed to assess the effect of treatment with a proteasome inhibitor (Bortezomib-Sigma-Aldrich, Burlington, MA, USA) on the morphology of *C. neoformans*. Bortezomib (BTZ—1.4 mg/kg, humanized dose) [43] was administered intraperitoneally 1 hour before infection. After 6 h and 10 days of infection, the animals were euthanized to obtain BALF and evaluate fungal morphology. The infected and untreated (NT) groups were used as controls.

### 2.4. Statistical Analysis

All statistical analyses were performed using GraphPad Prism, version 5.00, for Windows (GraphPad Software, San Diego, CA, USA), and the results were considered significant at *p* < 0.05. The survival curve was plotted by Kaplan–Meier analysis and the results were analyzed by the log rank test; for behavior parameters, the area under the curve was analyzed. Laccase, urease, morphology, phagocytosis, IPR assay, ROS and PRN measurements, and CFU per gram were analyzed by analysis of variance (ANOVA) followed by Tukey’s multiple comparison test. All correlation measurements used throughout this manuscript are Pearson correlations.

## 3. Results

### 3.1. Strain Growth and Morphological Characteristics in Culture

To study morphological and transcriptional responses to pulmonary infection, we focused on the serotype A strain H99 (VNI genotype) commonly used for evaluating virulence traits, and we selected four additional strains for comparison (Table 1). The additional strains included two of serotype A (WM148, genotype VNI and WM626, VNII), a serotype D strain (WM629, VNIV) and a serotype AD strain (WM628, VNIII). Initially, we evaluated the growth of the five strains in rich medium (Yeast Extract Peptone Dextrose (YPD)) and in minimal medium (MM) at 37 °C. All the strains demonstrated reduced growth in YPD compared to MM (Figure 1A–E). Results were confirmed by the plating method (data not shown). We also tested growth and melanin production after five days of culture in medium with the substrate L-DOPA. The WM628 and WM626 strains demonstrated a reduced melanization compared to H99, and WM626 additionally showed reduced growth (Figure 1F). An assay for laccase activity confirmed the visual analysis of melanin production (Figure 1G). Urease activity was also analyzed, but no significant difference was observed among the strains (data not shown). In the context of the growth studies, we next determined whether morphological differences were apparent. The strains were incubated in YPD or MM, and cell morphology was analyzed by microscopy (Figure 1H–K). Notably, we observed that all of the strains produced smaller cells upon growth in MM (Figure 1H,K). We also noted that the WM629 strain had a smaller cell body and a higher surface/volume ratio (S/V ratio) than H99 in YPD (Figure 1H,I). Under the growth conditions in YPD, a capsule was observed for WM629 and H99, and the size was larger for strain WM629 (Figure 1J). All of the strains produced a capsule in MM with the exception of WM148 (Figure 1J,K). Taken together, these results indicated conserved cell size responses to rich and minimal medium for the strains but revealed differences between the strains in the virulence traits of melanin and capsule production.

### 3.2. C. neoformans Strains Display Different Disease Profiles in Murine Cryptococcosis

After investigating growth and the in vitro virulence-related attributes, we evaluated whether these factors could be associated with the ability of each strain to cause disease in a murine model of cryptococcosis. Mice were infected intratracheally with cells grown in YPD, and survival, behavior, and fungal burden were evaluated. Differences in lethality were found for all strains (Figure 2A). Strains H99 (mean lethality time—MLt = 22 days) and WM629 (MLt = 23 days) caused an early lethality profile, while longer survival was observed for the WM628 (MLt = 28 days) and WM148 (MLt = 32 days) strains (Figure 2A). WM626 did not cause death and was classified as a non-lethal (NL) strain, although it induced behavioral changes at the beginning of the infection (Figure 2D–H). Thus, according to the survival data, the strains were classified as hypervirulent (H99, WM629 MLt = 21–25 days), virulent (WM628 MLt = 26–31 days), hypovirulent (WM148 MLt = 32–40 days), and non-lethal (WM626).

We next analyzed fungal burden in the two compartments of the lungs, considering the sequence of tissue colonization: the bronchoalveolar space (BS, obtained from bronchoalveolar lavage fluid) and the pulmonary epithelium (PE). Fungal burden recovered from the PE and bronchoalveolar lavage fluid was similar for the H99 and WM629 strains (Figure 2B,C). Compared to H99, a lower fungal burden was recovered from the bronchoalveolar space (BS) of mice infected with WM628 but not from the PE (Figure 2B,C). The hypovirulent WM148 and non-lethal WM626 strains presented lower fungal burdens in the PE and BS than H99 (Figure 2B,C).

The behavior of infected animals was analyzed using the SHIRPA protocol, which mimics human signs and symptoms caused by neurological diseases [44]. All the strains were able to influence the following parameters evaluated at ten days post infection: muscle tone and strength, neuropsychiatric status, autonomic function, motor behavior, and function and sensory reflex (Figure 2D–H). The hypervirulent and virulent strains showed more intense behavioral changes when compared to the hypovirulent strain after 22 days of infection (Figure 2D–H).

### 3.3. Hypervirulent and Virulent Strains Increased Fungal Burden in the Brain

As *C. neoformans* is associated with meningoencephalitis, we evaluated the fungal burden in the brain throughout infection for the strains with different virulence phenotypes. Unfortunately, we could not analyze fungal cell morphology directly from the brain tissue, although several protocols were tested. However, we verified that all strains were able to disseminate to the CNS after 6 or 24 h of intratracheal infection. In addition, the hypervirulent and virulent strains increased fungal burden over time, while the less virulent or non-lethal strains had a similar fungal burden in the CNS throughout the infection, with a lower fungal burden compared to H99 (Figure 2I).

Melanization is considered one of the main factors related to the tropism of *Cryptococcus* spp. to the CNS [16,18]. However, strains that were not able to produce melanin in vitro were able to translocate to the CNS and maintain a stable fungal burden. Therefore, we evaluated whether in vivo melanization was compatible with the in vitro observations for these strains. An intracranial mice infection was performed, and histological analysis of the brain was performed after ten days using Fontana–Masson stain. In the histological analysis, we observed cryptococcal cells with brown staining, characteristic of melanized cells. Interestingly, the strains that did not produce melanin in vitro (WM628 and WM626) were able to produce it in the brain (Figure 2J).

### 3.4. C. neoformans Hypervirulent Strains Are Less Susceptible to the Fungicidal Activity of Macrophages

In general, the evaluation of the ability of the strains to cause cryptococcosis did not strictly correlate with the elaboration of virulence traits. We therefore evaluated the interaction between the fungus and murine macrophages considering the importance of *C. neoformans* ability to survive inside the host’s immune cells. Phagocytosis did not vary according to the virulence of *C. neoformans* strains (Figure 2K). However, the virulent strains H99, WM629, and WM628 showed higher intracellular survival (Figure 2L) and were associated with an increase in intracellular levels of reactive oxygen species (ROS) and peroxynitrite (PRN) when compared to the H99 strain after 3 h of infection. However, the survival mechanism was overall independent of oxidative and nitrosative stress for the other strains (Figure 2M,N).

### 3.5. The Kinetics of In Vivo Morphology Revealed Small Cells at Early Stages of Infection with Later Capsule Enlargement

Considering the in vitro morphological response of *C. neoformans* to culture conditions, we next evaluated whether this variation in cell size occurred during murine lung infection and the possible relationship to virulence. Mice were infected with one representative strain of each virulence phenotype and euthanized after 6, 24, or 240 h of infection. Cell morphology was determined directly from the BS (BALF) and PE. *C. neoformans* cells were not detected in the brain by microscopy at these early stages of infection. We found that murine infection led to morphological changes that were related to the virulence of the strains. Specifically, all strains presented a smaller cell body (≤3 µm) in the BS and PE than the initial inoculum after 6 h of infection (Table 2), leading to a predominance of small cells with a higher S/V ratio (Figure 3A–F). Furthermore, the hypervirulent strain (H99) had a cell body 20% smaller in the PE (mean = 2.0 µm) when compared to the cells in the BS (Mean = 2.5 µm).

After 24 h, the hypervirulent (H99) and virulent (WM628) strains showed an increase in cell body size, accompanied by a lower S/V ratio in the BS (Table 2) (Figure 3A–F). However, the same increase did not occur in the PE, where we observed the predominance of smaller cells, thus providing evidence for tissue compartmentalization (Figure 3H). Furthermore, the relative capsule size increased for all strains in the BS and PE throughout the infection, but this increase was significantly higher for the virulent strains (H99, WM628, and WM148). Further, there is a positive correlation between capsule size and virulence in mice, but not for the in vitro data (Figure 3G). Notably, the NL strain did not increase in cell size after ten days of infection (Figure 3A–F). Interestingly, after ten days of infection, the cells returned to their original total diameter (capsule + cell body); however, they had a larger polysaccharide capsule and a smaller cell body (Figure 3I). Thus, these results highlight that before capsule enlargement, *C. neoformans* reproduce and generate daughter cells with a small size initially after infection. After that, cell body enlargement and capsule synthesis occur throughout the infection. However, some cells may present an intense enlargement of the cell body leading to the formation of titan cells (>10 µm), as seen for the hypervirulent strain in BL, after 24 h and 10 days of infection (Table 2). Cell morphology and the dynamics of the morphological variation were observed under an optical microscope after staining with India ink (Figure 3J). The scanning electron microscopy of the BALF was performed to examine the hypervirulent (H99) and NL (WM626) yeasts (blue) after 6 and 240 h (10 days) of infection (Figure 3K). Bronchoalveolar space cells and red blood cells are pictured in green and red, respectively (Figure 3K). A delay in re-growing the cell body was seen in low virulent strains, pointing to a reduced microbial fitness. We therefore focused our subsequent transcriptome analyses on the H99 strain during lung infection.

### 3.6. Dual RNA-Seq Analysis of C. neoformans and Mice Revealed Transcriptional Profiles of Early and Late Events of Infection

To evaluate the transcription changes underlying morphological and other responses during lung infection, we characterized the fungal and host transcriptomes for early and late stages of infection, considering the BS as the first compartment of colonization. For these studies, mice were infected with the H99 strain, and the BALF was obtained after 6 h or 10 days of infection to perform dual RNA-seq. Yeast cells grown in YPD (before infection) were used as a control. We analyzed the upregulated and downregulated genes after 6 h and 10 days of infection, compared to the inoculated yeast (YPD vs. 6 h and YPD vs. 10 d) and the differentially expressed genes (DEGs) between 6 h and 10 days of infection (designated BAL6h vs. BAL10d). Table 3 describes the RNA-seq statistics. The low mapping to the *C. neoformans* H99 reference genome in the infection libraries is consistent with dual RNA-seq analysis, as previously shown by other studies, ensuring the quality of our results [45,46]. These analyses revealed transcriptional profiles of early and late stages of infection, and the profiles shared between these two time points as well (Figure 4A). In the later stages of infection, 860 genes were modulated, with 488 being upregulated and 372 downregulated (YPD vs. 10 d). Moreover, 602 genes at 6 h and 540 genes at 10 d were uniquely expressed. The regulation of these genes represents the potential adaptation mechanisms used by *C. neoformans* in vivo, compared to the yeast cells found in the environment. During mice infection, there were 83 DEGs, with 81 upregulated and 2 downregulated genes in 10 d compared to 6 h (6 h vs. 10 d) (Figure 4A). *C. neoformans* DEGs are shown in Appendix A.

Functional classification of the differentially expressed transcripts was performed using FunCat (Functional Catalogue), GO (Gene Ontology), and KEGG (Kyoto Encyclopedia of Genes and Genomes). Nineteen biological processes were significantly enriched (YPD vs. 6 h, YPD vs. 10 d, and BAL6h vs. BAL10d) (Figure 4B–D). Biological processes such as protein binding, ribosomal proteins, ribosome biogenesis, and translation were upregulated after 6 h of infection compared to the yeast grown in YPD (Figure 4B) (Appendix A). Melanin and the polysaccharide capsule are virulence factors classically described for *C. neoformans* [15,17]. Interestingly, after 6 h of infection, we identified genes upregulated for these virulence factors (LAC2, CAP64, CAP2, and CAS3) (Appendix A). This is consistent with the increase in the polysaccharide capsule observed in the AS and PE after 10 days of infection and the presence of melanin in the cell wall in intracranial infection after 10 days. A gene for chromatin remodeling (SNF59) was also upregulated (Appendix A). Ten days post infection (YPD vs. 10 d, BAL6 h vs. BAL10 d), several transcripts related to respiration, electron transport and membrane-associated energy conservation, aerobic respiration, and splicing were upregulated (Figure 4C,D) (Appendix A).

### 3.7. Ribosomal Modulation Is Important for C. neoformans Adaptation in the Early Time of Infection

Transcripts for five hundred and thirty-two genes were upregulated in *C. neoformans* in the BS after 6 h of infection (Figure 4A). Seventy-five of them were involved in ribosomal regulation, specifically thirty-two related to the small subunit, forty-two to the large subunit, one (RPF2 CNAG_01187) to 60S pre-ribosomal formation, and one (UTP10 CNAG_04370) for 18S ribosomal RNA biogenesis (Figure 4E) (Appendix A). This finding is consistent with proliferation and the morphological analysis of *C. neoformans* throughout infection. Initially, we observed smaller and more metabolically active yeasts, compatible with the higher S/V ratio. Thus, the transcriptional response likely reflects the protein synthesis necessary for the reproduction and initial adaptation of the fungus.

### 3.8. Inositol Metabolism and Proteasome Modulation Are Involved in the Late Adaptation of C. neoformans during Infection

Previous studies have demonstrated that inositol catabolism is involved in the virulence of *C. neoformans* [47,48,49]. Consistent with these studies, our results showed the upregulation of genes related to inositol metabolism after 10 days of infection compared to YPD (YPD vs. BAL10d) and 6 h (BAL6h × BAL10d) (Appendix A). Transcripts for sixteen genes related to inositol metabolism were upregulated, including two related to myo-inositol oxygenase (MIO) that converts myo-inositol to D-glucuronic acid, a substrate of the pentose phosphate cycle and a component of the polysaccharide capsule (Figure 4F) (Appendix A). These findings are consistent with the increase in the relative capsule size observed for the most virulent strains after 10 days of infection.

The proteasome is important for the degradation of cellular proteins and is involved in synthesizing the polysaccharide capsule in *C. neoformans* [50,51,52]. Interestingly, the proteasome-related gene RPT3 (26S proteasome regulatory subunit—CNAG_03904) was upregulated after 10 days post infection, compared to 6 h (BAL6h vs. BAL10d) (Figure 4G). Considering these findings, we evaluated whether the proteasome inhibitor bortezomib affects the morphology of *C. neoformans* recovered from the BS. Yeast cells recovered after 6 h of infection from treated mice exhibited increased cell body diameter compared to the yeasts recovered from non-treated mice (Figure 4G), pointing to the importance of the proteasome for *C. neoformans* cell remodeling during infection. The treatment did not alter the morphology after 10 days of infection (data not shown).

### 3.9. Transcripts Related to Energy Production Are Differentially Modulated during Infection

Throughout infection, transcripts for 22 genes were differentially modulated in all comparisons (YPD vs. BAL6h, YPD vs. BAL10d, and BAL6h vs. BAL10d), involving different biological processes such as respiration, electron transport, membrane-associated energy conservation, metabolism of energy reserves, the pentose phosphate pathway, and electron transport (Figure 5A,B). Most transcripts (15) were downregulated at the beginning of infection and upregulated after 10 days (Figure 5A) (Appendix A). Oxidative phosphorylation, a pathway related to energy production, was the main enriched pathway (Figure 5C) (Appendix A).

### 3.10. The Mouse Transcriptional Response to the Early and Late Stages of Infection

The comparison of the in situ mice transcriptional profiles between 6 h and 10 d post *C. neoformans* infection allowed the identification of transcripts for 4575 genes involved in the early and late host responses. In particular, we found that 1375 genes were upregulated at 6 h and 1823 were upregulated at 10 d (Appendix A). To categorize the differentially expressed transcripts and characterize their functions, the ShinyGO v0.75, GO (Gene Ontology), and KEGG (Kyoto Encyclopedia of Genes and Genomes) classifications were used. Functional enrichment showed that at the early stage of infection (6 h), there is an over-representation of genes coding for proteins involved in phagocytosis, such as integrins and cell receptors (*CD18, MR, TLR2,* and *TLR4*), proteins related to lysosomes (*LAMP*, *LIMP*, *DMXL*, *WDR7*, and *NCOA7*), and components of the NF-kB signaling pathway (*LYN*, *VAV*, *BCAP*, *NFKB*, *NFAT*, *API*, and *CARMA1*) (Figure 6A,B) (Appendix A). The phagosome, lysosome, and B-cell signaling pathways were also enriched in response to *C. neoformans* in the early stage of the infection (Figure 6C) (Appendix A). These results are consistent with phagocytosis and cell-migration-related genes being important for the early mice response to infection (Appendix A). The connection with phagocytosis is further emphasized by our observation of the upregulation of transcripts for SOD1 (superoxide dismutase [Cu-Zn]—CNAG 01019) in *C. neoformans*, an important enzyme for oxidative stress resistance within phagocytes [53]. On the other hand, functional enrichment of the upregulated genes at 10 days (compared to 6 h) was over-represented with genes involved in cell morphogenesis/migration, components of the Phosphoinositide 3-kinase (PI3K)/Akt (PI3K-AKT) signaling pathways, focal adhesion, and inflammation and cytokine/chemokine pathways (Appendix A). The identification of these genes begins to build a view of the host molecular profile at the later (10 d) stage of cryptococcosis (Figure 6D,E). Notably, the host transcriptome response to *C. neoformans* resembled that of other infectious or inflammatory diseases, including rheumatoid arthritis, legionellosis, salmonella infection, tuberculosis, Chagas disease, and leishmaniasis (Figure 6F) (Appendix A).

As summarized in Figure 7, our results demonstrate the dynamics of cellular remodeling of *C. neoformans* during infection.

## 4. Discussion

*C. neoformans* is an important human pathogen related to cryptococcosis, a public health issue with a high mortality rate [54]. This fungus can be found as a free-living cell and as an intracellular pathogen, therefore it is exposed to several environmental pressures, such as temperature and pH changes, soil nutrient availability, phagocytosis by free-living amoebae, interaction with other microorganisms and plants, and exposure to UV radiation [55]. The exposure to these hostile environments may have contributed to the evolution of virulence factors that also support the survival and interaction with mammalian cells [55]. Despite knowledge about the virulence attributes of *C. neoformans*, the influence of cellular morphogenesis on virulence and disease outcome and the mechanisms involved in cell plasticity during infection remain poorly understood.

In this study, we combined in vitro and in vivo assays to examine differences in the virulence of a set of five *C. neoformans* strains. Using a murine model of cryptococcosis, we found that the lethality of each strain ranged from hypervirulent (H99 and WM629), to virulent (WM628), to hypovirulent (WM148), to non-lethal (WM626). These differences provided a useful set of strains for better understanding virulence attributes, such as the ability to survive inside macrophages, morphological changes in vivo, dissemination, and the ability to proliferate in the CNS. Initially, we determined that the strains generally had a smaller cell body and larger polysaccharide capsule when grown in MM, compared to growth in the YPD. Interestingly, after 6 h of mice lung infection, we observed a predominance of small cells in the BS compared to the initial size of the inoculated cells. Presumably, these smaller cells result from initial proliferation at the early time of infection. We refer to this event as cell remodeling, and it was observed in all strains independent of the virulence level. It is well established that fungal cells adopt different strategies during tissue invasion, such as the filamentation in *Candida albicans* [13,56]. In *C. neoformans*, the generation of small cells at the early stages of infection may be a determining factor for the invasion of the PE, since they may have an increased ability to cross barriers compared to the regular cells. This possibility is consistent with the tissue compartmentalization at 24 h post infection, in which smaller cells were detected in the PE. Moreover, smaller cells may be more active metabolically since they have a higher S/V ratio, facilitating the exchange of nutrients with the environment and increasing their fitness [21]. Considering that the lung environment is stressful, the initial cellular remodeling may be critical for optimizing *C. neoformans* energy expenditure and PE invasion.

The ability to cross the blood–brain barrier is essential for CNS infection [57]. The mechanisms involved in this process (e.g., a Trojan horse process) are well described, and it is likely that fungal cell size influences fungal dissemination, since small cells may be more readily phagocytosed, exhibit higher intracellular proliferation and survival [15], and may spread quickly from the lungs throughout the bloodstream to reach the CNS. In this study, the predominance of small cells in the PE at the beginning of the infection is likely contribute to pulmonary escape and early detection in the CNS. Furthermore, we speculate that these small cells (<3 µm) may be the precursors of the seed cells (<6 µm), as recently described [6]. Interestingly, the seed cells present changes on the cell surface that impair immunological response and favor the dissemination to extrapulmonary tissues [6]. Thus, our findings, in light of the previous studies, demonstrate how *C. neoformans* remodels itself in the lungs at the beginning of the infection and promotes the development of a systemic disease.

Despite our evidence, the role of small cells during *Cryptococcus* spp. infection remains poorly understood. A recent study demonstrated a relationship between small cells and virulence after culture under laboratory conditions [7]. However, in our morphological analyses in vivo, the predominance of small cells at the beginning of the infection was independent of the virulence profile. However, the ability to re-growth the cell body after 24 h and 10 days of infection can be associated with increased microbial fitness. Therefore, we believe that this cell type is essential for initial adaptation to the lung tissue and escape to other tissues, but it is not a virulence determinant. We assume that subsequent cellular events (capsule enlargement and cell body re-growth) determine the pathogenic potential of *C. neoformans.*

After the invasion of the CNS, *C. neoformans* has access to catecholamines that can be used to synthesize melanin, an important virulence factor. Therefore, the melanin phenotype and laccase activity observed in vitro may explain the higher fungal burden in the CNS found for the hypervirulent strains. However, this correlation is not absolute, because strain WM628 was not able to produce melanin in vitro but showed an increase in melanin and the capsule in vivo. Perhaps this is a compensatory mechanism to ensure the survival of the fungus in the CNS.

Deployment of the capsule, for example, is the most studied *C. neoformans* virulence factor. Several aspects regarding capsule structure, antigenic properties, and role in virulence have already been reported [17,18,58]. In our study, in vivo capsule enlargement was positively correlated with mice mortality and was associated with a higher fungal burden in the BS and PE 10 days post infection for the virulent strains. The absence of or reduction in the capsule facilitates the host’s defense response and disease outcome [16,17]. However, it is worth mentioning that in vitro capsule enlargement in MM alone did not predict virulence in vivo. These findings highlight in vivo morphology studies’ importance in establishing an association with virulence.

*C. neoformans* cellular remodeling is dynamic, and in our study varied according to the time and compartment of infection. Considering the chronological events of cellular morphogenesis during infection, remodeling strategies related to survival, tissue invasion, and fungal dissemination may be expected early, while mechanisms related to evasion of the immune response and colonization of the host, favoring fungal maintenance within the host may appear later. Based on these ideas, we performed dual RNA-seq analyses of fungal and host cells from the BS using the hypervirulent strain H99. Previous studies have also performed transcriptional analysis of *Cryptococcus* from the cerebrospinal fluid [6,48,59] and cell culture. However, in general, we did not observe a similar transcriptional response comparing these studies. This difference may be related to differences in the fungal response to host niches. Here, we analyzed yeasts directly from the bronchoalveolar space, a totally different environment compared to cerebrospinal fluid and cell culture in laboratory conditions.

We initially observed that transcripts for many genes encoding ribosomal subunits were upregulated early after infection, a finding consistent with the adaptation to the host and the initiation of proliferation. That is, cells with higher ribosomal expression would show a higher growth rate [60]. This information aligns with our hypothesis that small cells from the BS show increased cell replication early in the infection, contributing to tissue invasion. Previous reports demonstrated alterations in ribosomal modulation in *C. neoformans*; however, little is known about the in vivo regulation of protein biosynthesis in *C. neoformans* [61,62,63]. Our results demonstrate this modulation in a murine model, considering early and late stages of infection, pointing to the importance in cell remodeling. This reinforces the importance of ribosomal regulation during infection, opening perspectives for further evaluation of its role in virulence and the search for new targets for antifungals.

We also found that transcripts encoding functions for energy metabolism were modulated throughout the infection. For example, a downregulation of transcripts involved in ATP synthesis was observed in the early stage of infection, possibly because cells were still adapting to the host from the pre-culture in a rich medium (YPD). We did note increased HXS1 (CNAG_03772) transcripts at the early stage, and this high-affinity glucose transporter is regulated by glucose and involved in fungal virulence [64]. In this context, the level of glucose in the BS seems to possibly be an early signal for the adaptation of *C. neoformans*. This idea is consistent with the expression of monosaccharide transporters previously observed during murine lung colonization [23]. Consequently, this regulation may contribute to cellular remodeling of *C. neoformans*, favoring the optimization of energy expenditure and increase in reproductive fitness. However, increased levels of transcripts involved in ATP synthesis in mitochondria were detected later in the infection, reinforcing previously published data that pointed to the importance of this organelle in fungal virulence [65,66,67,68]. This shift in the transcriptional profile occurred concomitantly with the increase in polysaccharide capsule, a high-energy-requiring process. Furthermore, we observed an increase in the regulation of insulin secretion in mice, corroborating previous data [46]. Therefore, this would possibly lead to lower glucose levels and stimulate the regulation of pathways for obtaining energy in *C. neoformans*. Thus, our findings reinforce the idea that *C. neoformans* needs to cope with a variety of stresses during infection (temperature, phagocytosis, oxidative and nitrosative stress, hypoxia, and low glucose levels) that activate energy-dependent resistance mechanisms (polysaccharide capsule synthesis, inositol, and proteasome regulation), which are critical for fungal survival.

Our RNA-seq analysis revealed additional features of fungal adaptation, including the upregulation of transcripts coding for proteins from the inositol pathway 10 days post infection, consistent with a previous study [48]. This pathway contains three genes that encode myo-inositol oxygenases, which convert myo-inositol into d-glucuronic acid, a substrate of the pentose phosphate cycle and a component of the polysaccharide capsule [48]. Furthermore, the upregulation of this pathway is consistent with increased polysaccharide capsule and higher fungal replication in the CNS, since inositol plays a role in *C. neoformans* brain invasion [49]. We also found that the transcript of the RPT3 gene (CNAG_03904) encoding a putative subunit of the proteasome 2 was upregulated 10 days post infection. The proteasome is essential in cell cycle regulation, transcription, signal transduction, apoptosis, and polysaccharide capsule synthesis [50,52]. The importance of the proteasome is supported by our observation that treatment of infected mice with a proteasome inhibitor (bortezomib) impaired *C. neoformans* cell remodeling after 6 h of infection. In addition, a previous study demonstrated that bortezomib inhibited the polysaccharide capsule synthesis in vitro [52]. However, this does not exclude the possibility that BTZ also affected the mouse proteasome, or another pathway, contributing to the observed effect on fungal cell size. These findings reinforce the role of the proteasome in *C. neoformans* cell remodeling and generate perspectives for the repositioning of bortezomib and other proteasome inhibitors for the treatment of cryptococcosis. Although we did not observe morphological differences after 10 days of treatment (data not shown), this does not exclude the possibility that the treatment induces physiological changes in the fungal cell, such as changes in cell viability and replicative rate. We believe that proteasome assembly is important throughout the infection, and it may help the fungus to cope with the strong immune response against it. However, further studies are still needed to better understand the effect of bortezomib and the proteasome on late times of *C. neoformans* infection.

Considering the mouse response to infection by *C. neoformans*, the most highly upregulated transcripts after 6 h of infection were involved in recognition, uptake, and phagocytosis, including IL-18 and TNF. IL-18 is critical in TH1 responses via IFN-y production, while TNF is associated with activating innate and adaptive responses against *C. neoformans* [69,70]. After phagocytosis, the fungal cell finds an environment with reduced oxygen levels, explaining the downregulation of COX1 and COX2 genes in *C. neoformans*. These genes are involved in Complex IV of the electron transport chain, and their activity requires oxygen as the final electron acceptor [71]. Consistent with our results, previous findings also demonstrated that the in vitro macrophage-induced hypoxia environment leads to the downregulation of COX1 regulation in *C. neoformans* [71]. However, the mouse response is ineffective in controlling *C. neoformans* growth, as evidenced by the higher fungal burden in the organs after 10 days post infection. We hypothesize that small cells may be more readily engulfed by macrophages. However, a higher fungal reproductive fitness of small cells may increase the capability for intracellular proliferation, favoring the selection of these cells to facilitate persistence and disease. Although the timing of the host response plays a role in determining the extent of tissue damage [72], it is ineffective in fungal control. It is unclear how the dynamic response differs for the hypovirulent and non-virulent strains, and this is an important area for future study. Still, we hypothesize that the balance of the pro-inflammatory and anti-inflammatory responses added to immunomodulatory mechanisms (polysaccharide capsule) [73] may be decisive. Since non-lethal and hypovirulent strains have a reduced capsule at late times of infection, this may contribute to the maintenance of a more effective inflammatory response.

## 5. Conclusions

In conclusion, our findings support a scenario in which the fungus initially deploys mechanisms aimed at survival, proliferation, and tissue invasion. Later, the fungus may focus on mechanisms aimed at resistance to the immune response and colonization, which are crucial for disease development. A limitation of this study is that it used only four strains of *C. neoformans* to evaluate fungal cell remodeling in vivo. As most of the experiments were conducted in a murine model, this makes the use of many strains unfeasible. In fact, how the differences shown here work for other lineages of the *C. neoformans* complex still need to be elucidated. Despite this, the study brings important contributions on the dynamics of cellular remodeling of *C. neoformans* during infection and generates perspectives for the survey of new therapeutic and diagnostic targets for cryptococcosis.

## Figures and Tables

**Figure 1 cells-11-03896-f001:**
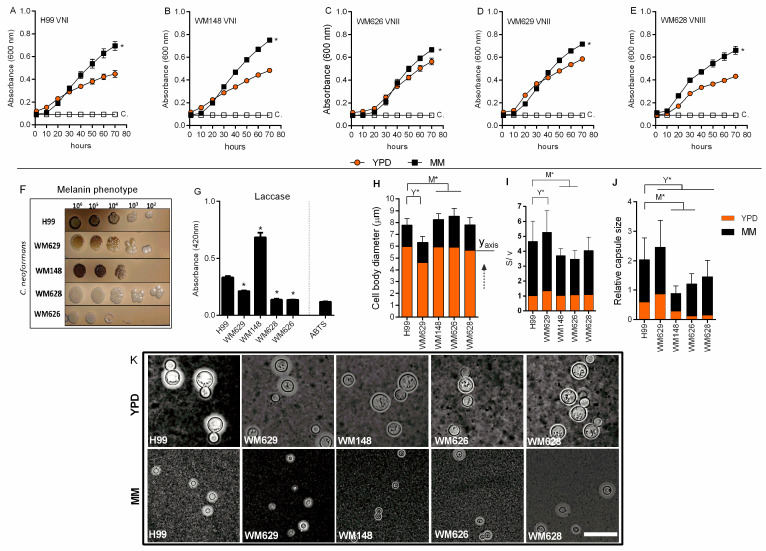
Phenotypic analysis of *C. neoformans* strains in vitro. (**A**–**E**) Growth curves of each strain in Yeast Extract Peptone Dextrose (YPD) or in minimal medium (MM) at 37 °C (blank squares represent only the culture media). (**F**) Visual analysis of melanin production after fungal growth in solid minimal medium with L-DOPA at 37 °C for 5 days. (**G**) Laccase activity measured in MM (ABTS: 2,2′-azinobis(3-ethylbenzthiazoline-6-sulfonate). (**H**) Cell body diameter in YPD or in MM at 37 °C after 72 h. Dashed arrow: indicates that the MM y-axis starts at the end of each YPD bar. (**I**) Surface/volume ratio in YPD or in MM at 37 °C after 72 h. (**J**) Relative capsule sizes in YPD or in MM at 37 °C. Mean values were compared with H99 (* *p* < 0.05; M* = strain analysis on MM; Y* = strain analysis on YPD). (**K**) India ink counter-staining of *C. neoformans* after growth in YPD or MM. Scale bar: 10 μm.

**Figure 2 cells-11-03896-f002:**
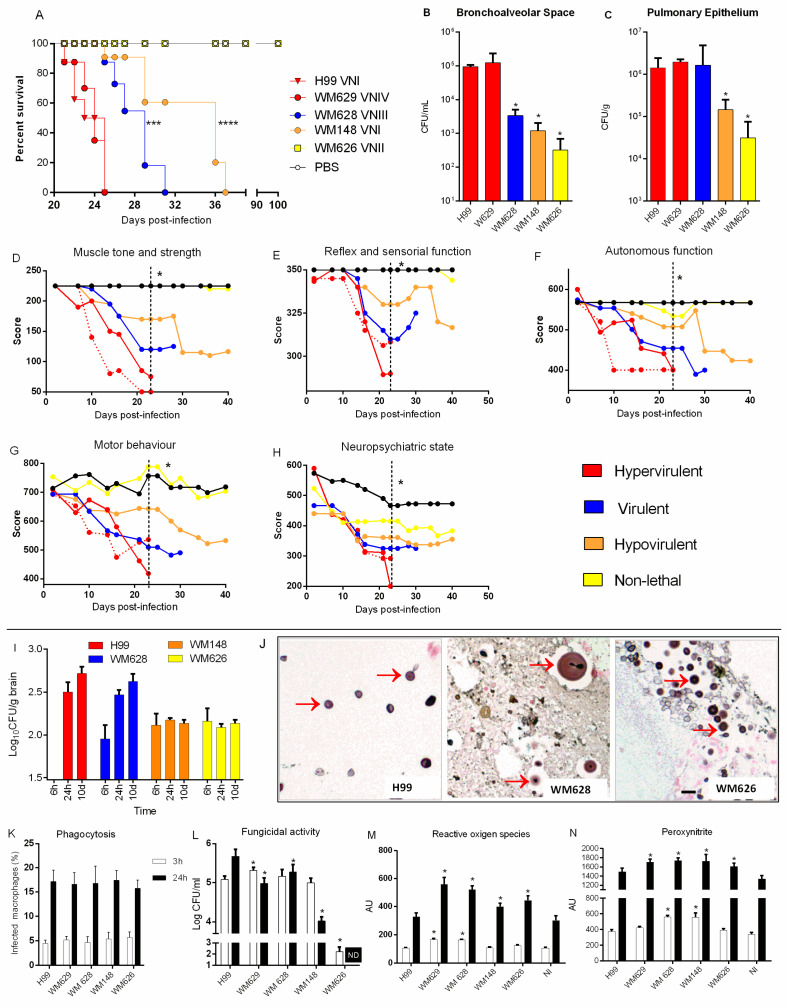
Virulence analysis of *C. neoformans* strains in vivo and ex vivo. (**A**) Mortality curve of mice after intratracheal infection with 1 × 10^5^ cells (*** *p* < 0.001 and **** *p* < 0.001). (**B**,**C**) Fungal burden determination from the bronchoalveolar space (BS) and pulmonary epithelium (PE) 10 days post infection. (**D**–**H**) Behavior analysis after infection with *C. neoformans* strains. Dashed lines indicate the statistical difference in virulent, hypovirulent, and non-lethal strains compared with H99 (* *p* < 0.05). Virulence was classified according to the mean lethality time (MLt), with hypervirulent (red) range of MLt = 21–25 days, virulent (blue) range of MLt = 26–31 days, hypovirulent (orange) range of MLt = 32–40 days, and non-lethal (yellow) range of MLt => 100 days. (**I**) Fungal burden in the brain after 6 h, 24 h, and 10 days of intratracheal infection. (**J**) Brain stained with Fontana–Masson after 10 days of intracranial infection. Red arrow indicates the stained fungus due to the presence of melanin in the cell wall. Scale bar: 10 μm. (**K**) Phagocytic index after 3 h and 24 h infection. (**L**) Colony forming unit (CFU) determination represents viable yeast cells internalized by macrophages in each time point. (**M**) ROS and (**N**) PRN production after 3 and 24 h of infection by *C. neoformans*. Mean values were compared with H99 (* *p* < 0.05). AU: arbitrary units of fluorescence. NI: non-infected macrophages.

**Figure 3 cells-11-03896-f003:**
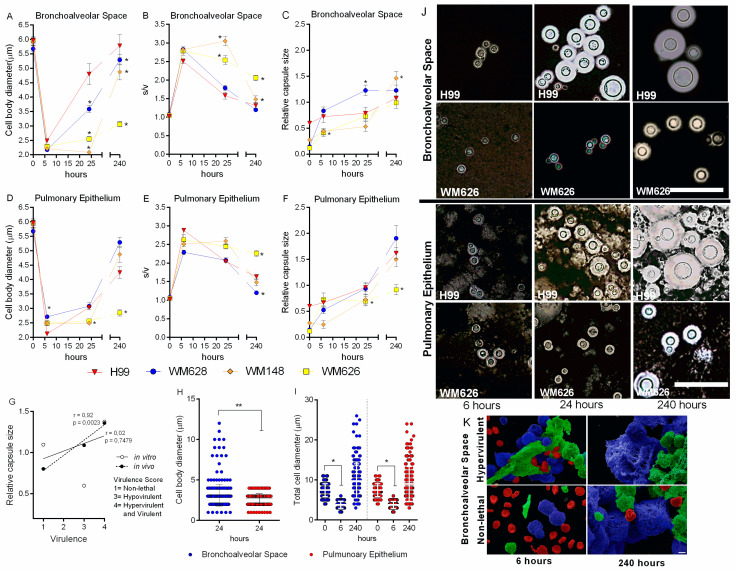
Dynamics of morphological variation of *C. neoformans* in vivo. (**A**–**F**) Morphological variation in bronchoalveolar space (BS) and pulmonary epithelium (PE) of four strains of *C. neoformans* after 6, 24, and 240 h of intratracheal infection. Mean values were compared with H99 (* *p* < 0.05). (**G**) Correlation between virulence and relative capsule size after 240 h infection. R values are included next to lines of best fit and are based on Pearson correlation. (**H**) Cell body diameter in BS and PE after 24 h of infection. Mean values were compared with bronchoalveolar space (** *p* < 0.01). (**I**) Total cell diameter in BS and PE after 0 and 240 h infection. Mean values were compared with 0 h of infection (* *p* < 0.05). (**J**) India ink counter-staining of the yeasts recovered from infected mice. Scale bar: 10 μm. (**K**) Scanning electron microscopy (SEM) of hypervirulent and non-lethal strains after infection: yeasts are colored in blue, bronchoalveolar space cells in green, and blood cells in red. Scale bar: 2 μm.

**Figure 4 cells-11-03896-f004:**
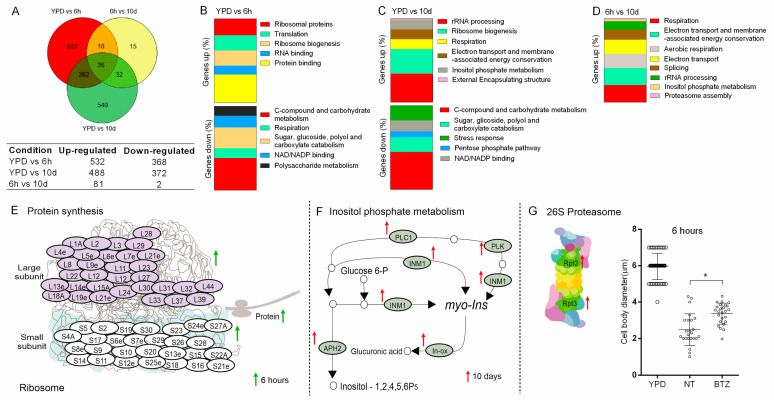
Differentially expressed genes (DEGs) in *C. neoformans* strain H99 during mice infection. (**A**) Venn diagram showing the number of H99 transcripts modulated comparing 6 h and 10 d of infection and YPD cultured yeasts. (**B**–**D**) GO biological processes enriched in H99 after 6 h and 10 days of intratracheal infection, for each comparison. (**E**) Upregulated transcripts related to ribosome subunits after 6 h of infection compared to YPD. (**F**) Genes involved in inositol metabolism differentially expressed after 10 days of infection (CNAG_00038, CNAG_00730, CNAG_00869, CNAG_01823, CNAG_01947, CNAG_02867, CNAG_04209, CNAG_04335, CNAG_04869, CNAG_05316, CNAG_05884, CNAG_06348, CNAG_06623, CNAG_06785, CNAG_06967, and CNAG_07799). (**G**) Differentially expressed genes related to fungal proteasome 10 d post infection, and the influence of bortezomib (BTZ) (proteasome inhibitor) on cell body diameter. Mean values were compared with NT (* *p* < 0.05). Upward arrows indicate induction of gene expression. Upregulated genes after 6 h are indicated by green arrows and upregulated genes after 10 days are indicated by red arrows.

**Figure 5 cells-11-03896-f005:**
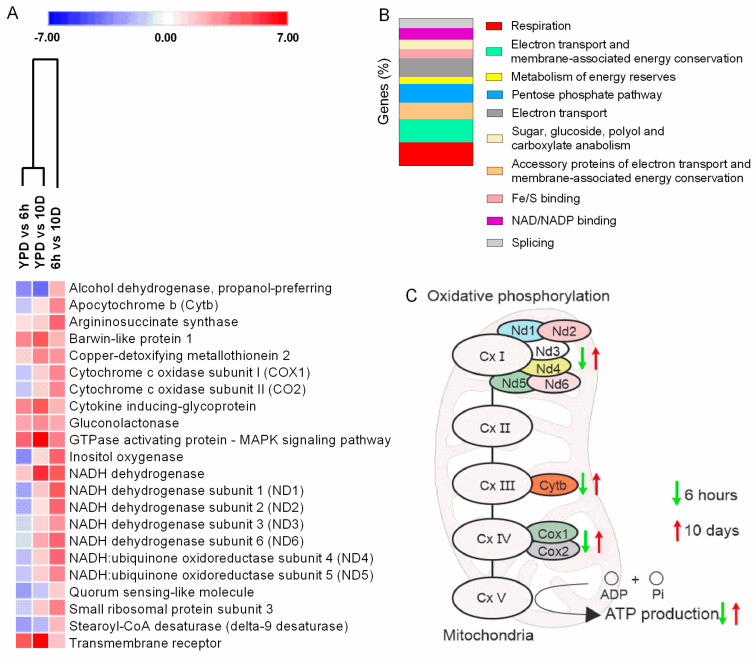
Transcriptional profile of *C. neoformans* in the bronchoalveolar space of mice. (**A**) Heatmap of the significantly enriched GO categories of regulated fungal genes at 6 h and 10 days post infection. (**B**) Enriched GO biological processes of fungal genes expressed in the bronchoalveolar space. (**C**) Modulated genes from the oxidative phosphorylation pathway expressed 6 h and 10 days post infection. Upward arrows indicate induction of gene expression and downward arrows indicate repression of gene expression at 10 days, compared to 6 h. Differently regulated genes after 6 h are indicated by green arrows, and differentially regulated genes after 10 days are indicated by red arrows.

**Figure 6 cells-11-03896-f006:**
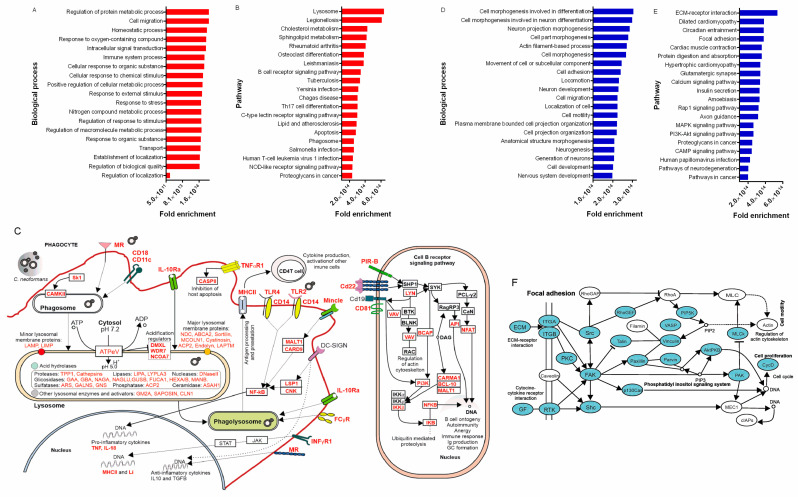
In situ transcriptional profile in the early and late mouse response to *C. neoformans* infection. Genes are upregulated at 6 h compared to 10 d. (**A**) Biological processes and (**B**) pathways enriched in the BS of mice 6 h post infection. (**C**) Gene regulation of phagocytosis and B-cell receptor signaling in the BS. After deposition in the lungs, *C. neoformans* can be recognized by pattern recognition receptors (MR, CD18, CD11C, DC-SIGN, TLR4, and TLR2) on phagocytes and internalized, leading to the formation of the phagosome and consequently the phagolysosome. The acidification of this compartment combined with respiratory burst and degradative enzymes are important factors for antimicrobial activity. After the degradation of *C. neoformans* in the phagolysosome, microbial antigens can be targeted to the MHC-II pathway for presentation to CD4 T cells. Furthermore, cell wall components of *C. neoformans* (GXM, GalGXM, and mannoproteins) can also be recognized by B-cell receptors, leading to signaling for antibody production. Upregulated genes in red. Based on KEGG and Gene Ontology (GO) analyses. Genes are upregulated at 10 days compared to 6 h. (**D**) Biological processes and (**E**) pathways enriched in the BS of mice 10 days post infection. (**F**) Focal adhesion pathway showing the upregulated genes (blue) in the BS 10 days post infection. Based on KEGG and Gene Ontology (GO) analyses.

**Figure 7 cells-11-03896-f007:**
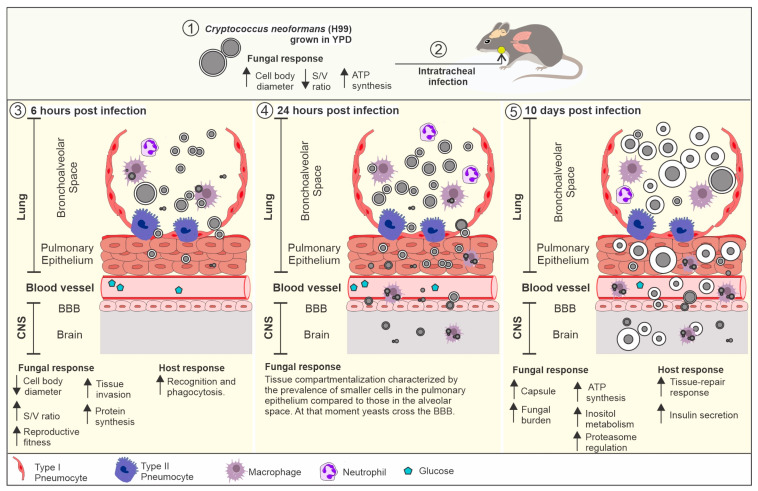
The dynamics of *Cryptococcus neoformans* cell remodeling during infection. (**1**) The growth of the H99 strain on YPD resulted in cells with a larger cell body diameter (mean: 5.6 µm), reduced capsule (mean: 0.5), reduced surface-to-volume ratio (mean: 1.16), and upregulation of pathways related to energy production. This profile mimics yeasts found in the environment. (**2**) After mice intratracheal infection with these cells, we observed a time- and location-dependent cell remodeling. (**3**) At 6 h post infection, we observed a predominance of small cells in the BS compared to the initial size. These cells have a smaller cell body and a higher surface-to-volume ratio. This contributes to increased fitness and pulmonary epithelium invasion. This cellular profile was associated with upregulation of pathways related to protein synthesis and upregulations of recognition and phagocytosis pathways in the host. (**4**) At 24 h post infection, we observed tissue compartmentalization characterized by the prevalence of smaller cells in the pulmonary epithelium (variation: 1.0–4.0 µm) compared to cells in the BS (variation: 1.0–12.0 µm). This evidence confirmed our hypothesis that small cells are important for PE invasion during interaction with the host. At this stage, the cells were able to reach the bloodstream, free or internalized by phagocytes, and cross the blood–brain barrier (BBB) and infect the central nervous system (CNS). (**5**) At 10 days post infection, we observed an increase in the polysaccharide capsule and upregulation of pathways related to energy production, inositol, and proteasome metabolism. In the host, this cellular profile led to the upregulation of tissue repair pathways and insulin secretion.

**Table 1 cells-11-03896-t001:** *C. neoformans* strains used in this study (American Type Culture Collection).

Strain	Genotype/Serotype	Isolation Source	Origin
WM 148 (ATCC^®^ MYA-4564^™^)	VNI A	CSF	Clinical–Australia
WM 626 (ATCC^®^ MYA-4565^™^)	VNII A	CSF	Clinical–Australia
WM 628 (ATCC^®^ MYA-4566^™^)	VNIII AD	CSF	Clinical–Australia
WM 629 (ATCC^®^ MYA-4567^™^)	VNIV D	Blood	Clinical–Australia
H99 (ATCC^®^ 208821^™^)	VNI–A	CSF	Clinical–North Carolina-USA

CSF: cerebrospinal fluid.

**Table 2 cells-11-03896-t002:** Cell body diameter variation in *C. neoformans* in vivo.

Cell Body Diameter (µm)	Cells (%)							
	Bronchoalveolar Space 6 h				Pulmonary Epithelium 6 h			
	H99	WM628	WM148	WM626	H99	WM148	WM628	WM626
**≤3**	100	100	100	100	100	100	100	100
**4–5**	0	0	0	0	0	0	0	0
**6–9**	0	0	0	0	0	0	0	0
**>10**	0	0	0	0	0	0	0	0
	**24 h**				**24 h**			
**≤3**	46.3	43.4	96.7	100	84.5	82.6	95.8	96.3
**4–5**	25.9	56.6	3.3	0	15.5	11.5	4.2	3.7
**6–9**	18.5	0	0	0	0	5.9	0	0
**>10**	9.2	0	0	0	0	0	0	0
	**240 h (10 days)**				**240 h (10 days)**			
**≤3**	20.3	7.8	33.3	76.1	46.2	3.9	37.1	84.7
**4–5**	33.5	61.5	24.0	22.6	38.7	63.3	22.2	13.4
**6–9**	34.6	30.7	42.7	1.3	15.1	32.6	40.7	1.9
**>10**	11.8	0	0	0	0	0	0	0

**Table 3 cells-11-03896-t003:** Dual RNA-seq statistics of mapping the reads to the reference genome.

			*C. neoformans*		Mouse	
Sample	Raw Reads	High-Quality Reads	Mapped Reads	Total Mapped Reads (%)	Mapped Reads	Total Mapped Reads (%)
BAL10d I	35,858,545	34,810,495	415,287	1.19	20,978,194	60.26
BAL10d II	36,346,653	35,320,291	579,106	1.64	20,921,124	59.23
BAL6h I	34,773,919	33,841,813	283,360	0.84	20,098,380	59.39
BAL6h II	45,536,641	44,382,642	650,886	1.47	24,160,652	54.44
YPD IC	26,632,962	26,025,358	25,820,614	99.21	-	-
YPD2C	25,955,261	25,380,479	25,185,198	99.23	-	-

## Data Availability

Datasets were submitted to the Gene Expression Omnibus database (http://www.ncbi.nlm.nih.gov/geo/, accessed on 1 November 2022), accession number GSE206758.

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
