# Peer review of "The Dynamics of *Cryptococcus neoformans* Cell and Transcriptional Remodeling during Infection"

_cells, 2022, doi:10.3390/cells11233896_

Round 1

Reviewer 1 Report

In the manuscript by Freitas G. J. C. et al., the Authors investigate cell size and capsule size and changes in gene expression in C. neoformans/C. deneoformans depending on the stage of infection based on a murine infection model. Initially, the Authors compare proliferation rate, cell and capsule size, as well as melanin production of five strains (C. neoformans WM148, WM626, C. deneoformans strain WM629, and a C. neoformans/C. deneoformans hybrid strain WM628), in two media types (YPD and a minimal medium). Subsequently, all five strains are tested in murine infection model, where fungal burden is measured in bronchoalveolar space (BS) and pulmonary epithelium (PE), and subsequently also in the brain (for 4 strains representing hypervirulent, virulent, hypovirulent, and non-lethal phenotypes). These studies are complemented with the investigations of phagocytosis and survival in murine macrophages. In addition, the Authors evaluated the behavior of infected animals using the SHIRPA protocol, which mimics the effects of human neurological diseases. Major findings are the report of relatively small cells during the initial stages of infection and a comprehensive investigation of C. neoformans strain H99 and the host transcriptomes at 6 hours and 10 days post infection. Overall, this extensive study contributes to a better understanding of gene expression changes in the pathogen and the host during infection. Below I posted specific comments that if followed may help to improve the manuscript. 

1.     The Authors demonstrate that during initial infection cell size is relatively small. The Authors point to the importance of a small cell size for the infection. However, there is no clear direct evidence from this study that small cells are better suited for infection. 

2.     Ln 15: neoformans should be italic

3.     Ln 18-19: The Authors refer to changes in cell size cellular remodeling. I am not sure this is a correct way to refer to it. What specific remodeling would the cell size change involve?

4.     Ln 18: the Authors refer to transcriptional factors not transcription factors - is this because the Authors don't necessarily refer to the TFs? Later in the text it becomes clear that what the Authors mean here is expression of protein-encoding genes in general. 

5.     Ln 20: perhaps the Authors should specify already in the Abstract how much later (the exact of approximate timing).

6.     Ln 21-22: Inhibition of what (the proteosome?) should be specified.

7.     Ln 65-67: the sentence should be rewritten.

8.     Ln 72: What are molecular signatures?

9.     Ln 86: 5 in 105 should be superscript

10.  Ln 114: superscript in 105

11.  Ln 202: superscript in 102

12.  Ln 252 and other parts: a concern that the Bortezomib may affect both the fungal cells and the host (it is an anticancer drug).

13.  In Figure 1, the A-E panel could be all labeled as panel A as each graph is distinctly labeled as a different strain.

14.  Ln 274-275: The results of the growth assay in vitro are surprising for two reasons: 1. Generally, one would expect cells in YPD to grow faster as compared to those incubated in the MM medium, at least initially as it is a rich medium. 2. cultures grown in YPD for nearly 3 days only reached OD600 close to 0.6, whereas typically one would expect the OD600 to be well above 1.0 at that time. What has accounted for such unexpected results? Is the fact that the cells are grown at 37C contributing to this unexpected inhibition of growth when cells are incubated in YPD?

15.  Ln 289-290: the results regarding cell size are not surprising and not novel.

16.  Ln 338-339: this is an interesting finding. What could account for this difference? Why despite melanization and the presence in the brain, strain WM 626 is not lethal. Perhaps the very low survival in macrophages accounts for that?

17.  Figure 2, panels M and N: what are the last sets of data in each graph (labeled AE if I read it correctly)?

18.  Ln 350-352: What cells were the ROS and PRN measured in? - the fungal cells or the macrophages?

19.  In Figure 3K, other elements (colored in red and purple) should also be described.

20.  In the data analysis for the RNAseq experiment, perhaps it would be better to utilize the culture that was used for inoculation as a control Instead of YPD 6 and YPD 10d.

21.  For the RNAseq study, what phase of growth were the cells utilized for the inoculum? Was the inoculum a stationary phase cell culture or exponentially grown cells? This would influence the transcriptional profile.

22.  Ln 502-504: Instructional sentence needs to be deleted

23.  Ln 688-689: should be .. the ability to proliferate in in the CNS

24.  Ln 690: compared to growth in the YPD

25.  Ln 717-718: this is an interesting finding (should be emphasized in the Abstract)

26.  Ln 724-725: The Authors hypothesize that small size during initial infection is a way to adapt to the lung tissue. Notably, small size is also characteristic of cells incubated in MM medium. Perhaps lung environment is also nutrient poor and therefore leads to similar cell size as MM.

27.  Ln 755: ...data is needed to explore these functions...

Reviewer 2 Report

In Freitas et al the authors are addressing the question whether there are morphological and transcriptional changes over time during an infection with Cryptococcus neoformans. This is actually well known, and has been a topic of intense study. The authors claimed that most studies have been in vitro, and that in vivo studies are lacking. That is not completely true, as several studies in the early 2000 reported phenotypic changes occurring in vivo: Fries et al 2001 and 2002, and Charlier et al 2005, for example. More recently, small cells have been shown to be better at dissemination (Denham et al 2022) and to survive better intracellularly (Hommel et al 2019). Only one of those papers was cited, and none were discussed, which feels misleading. In that sense, this article is not contributing anything innovative. The lab of John Perfect has also analyzed transcriptional changes on the fungus directly from human CSF infections twice (Chen et al 2014 and Yu et al 2021), and others have done it from in vitro cultures (Casadevall lab, Nelson lab, Wozniak lab). The least the authors could do is compare their mouse findings to the human findings and/or the in vitro findings. They completely ignore those other reports. However, they did a massive amount of work trying to focus on in vivo changes, and although others have done that too, this manuscript has some interesting findings that should be published in light of all other findings. Based on their results, they did a good job summarizing and writing the paper, but terrible use of citations. Their methodology/approach was sound, and the use of dual-RNASeq I think is original (not sure). Below are my concerns/comments and I leave it to editor to decide how many, if any, need to be addressed before reconsideration for publication. At the very least, I would ask the authors to acknowledge the other work mentioned above.

Main concern: other important and relevant work on the same topic is ignored, not cited nor discussed. Also, some of their conclusions overstate their results, although if taken together with other published results, their conclusions make sense. The problem is that they do not cite the other work! For example, in line 646 they say “This evidence confirmed our hypothesis that small cells are important for tissue invasion…” What is the evidence? Did they look at blood burden, or other tissues? If you restrict your analysis to the results presented solely on this manuscript, this statement is not supported. They state that at 24 hr, when they see small cells in the PE, that’s when the fungus reached the brain. However, in Fig. 1I they show that even at 6hr there is significant burden in the brain. However, if you take into consideration all the other published data, especially the paper characterizing “seed cells” then their statement makes sense. They just have to acknowledge those other papers and discuss them.

Minor concerns:

1.      Line 34 – they cite the 2017 paper. These figures have been updated now, they should cite the new paper (Lancet Infect Dis. 2022 doi: 10.1016/S1473-3099(22)00499-6).

2.      Line 80 – what is Minimal Media? Nowhere on the paper they state the recipe. It is important to know in order to compare to other results. Specially because that media was also used to induce capsule, which typically is done using host-like conditions such as DMEM, 5% CO2, and 37°C (see my note below about that).

3.      For Fig. 1A – E, the size differences are so great between YPD and MM, that you cannot use OD600 to compare growth. They say the same results were seen by CFUs, but do not show it. When comparing particles that differ so much in size, OD is not a good comparison. Also, they should say in the legend what is the blank squares in the graphs (I’m assuming it is just media).

4.      For Fig. 1H – J the graphs are a bit complicated. At first glance, it looks like the MM are bigger, since they go all the way to 8 and 9 um. Then in the text they say they MM are smaller, then I noticed the y-axis. Stacked bars might save space but are harder to read. Bar graphs are easier and will show the difference better.

5.      For Fig. 1K, why use MM instead of actual capsule-inducing media? Unless MM is a known capsule-inducing media, but the reader would not know since MM is not defined.

6.      For Fig. 2A, they state that WM626 is non-lethal, but only show data up to day 37. In the legend they say that the MLt was >100 days, then why not show that in the graph with a cut axis? Also, WM626 is sometimes called non-lethal and sometimes non-virulent, it should be called non-lethal or attenuated, but it is clearly virulent because it disseminates and colonizes the brain, although fungal organ burden at the termination of the experiment (like day 37, for example) would solve that issue.

7.       The heading 3.3 (line 323) is misleading, all strains were able to reach the brain, but the heading says that only hypervirulent and virulent strains colonize the brain.

8.      Fig. 2M and N have an extra set of bars on the far right labeled AE. What is AE?

9.      Lines 367-368 – the symbol for micrometers appears as a “@”.

10.   Fig. 3K is never mentioned in the text. Also, in the Fig. legend is not stated what’s purple? Are those immune cells or large, encapsulated cryptococcal cells?

11.   In Fig. 4F, glucuronic acid is called glucoronate. For simplicity, the authors should be consistent.

12.   For Fig. 4G, the authors tested an effect of bortezomib at 6 hr, but they say in the text that the change in proteasome genes was seen at 10 days, not 6 hr, so why looked at 6 hr? They saw a change, but it is unclear what it means given that they did not see a change in gene expression at 6 hr. Also, they suggest in their discussion that bortezomib could be used as an antifungal, but that would also affect the host. How will they obtain specificity against the fungus and not the host? They do not discuss that.

13.   Lines 502-505 are leftover instructions. Please delete.

Reviewer 3 Report

In this study the authors examined the cellular remodeling of Cryptococcus neoformans (Cn) during growth and infection. Five strains/isolates of Cn were used to examine growth, capsule size melanin production and virulence in vitro and/or in vivo. The authors went on to perform transcriptomics of H99 (one of the strains of Cn) and on the host. Although the study is largely descriptive, it is a comprehensive analysis of the cellular remodeling/ responses of Cn during infection. Overall the study is rigorous, and well-done. The conclusions stated are supported by the data presented.  

The major limitation of the study is that only 4 other strains in addition to H99 (the common laboratory strain) were used in this study and that they all originate from Australia. It would have been more interesting to perhaps examine and compare clinical isolates from additional regions/countries. The authors are aware of this limitation and mention this in the conclusion. Despite this limitation, however, the study does provide insight into the dynamics of the infectious process from the bronchoalveolar space to the CNS.

The following comments should be addressed:

1.     What is the history and/or information regarding the Cn strains used in this study? Are they clinical isolates taken from patients and if so what was the status of the patient (i.e., presence of CNS involvement?)

2.     For the transcriptomics studies, why was growth of Cn in YPD used as a baseline/control instead of growth in minimal media? How would the profile change if the comparison was against growth in minimal media?

3.     There is no mention of any inflammation or markers of inflammation 10 days post-infection?

4.     Why was an inoculum of 105 chosen?

5.     Only male mice were used in this study but the reason for this is not justified/discussed. Would you anticipate a different result in female mice? Is sex a variable here?

6.     The authors state at the top of 3.3 that, “…we verified that all strain were able to disseminate to the CNS after 6 or 24 h of intratracheal infection.” Doesn’t this suggest that the size of the cell body of Cn or the capsule size, which are different at 6 and 24 h (Figure 3) does not influence the ability of Cn to enter the CNS? Is entry into the CNS independent of “size”?

7.     Related to point #6 – in Figure 2I, no fungal burden is reported for H99 at 6h? This contradicts the statements made previously.

Round 2

Reviewer 1 Report

The Authors addressed most comments/issues, which resulted in improved manuscript. I have three follow-up new comments that I believe should be still addressed/followed before the manuscript is accepted. 

1. My new comment to the response to the initial comment #8: I suggest the Authors actually replace the "molecular signatures" with "transcriptional profile" - that way the reader will understand that here the Authors refer to transcriptional profile and not to some other (ill-defined) feature.

2. My new comment to the response to comment #12: I suggest the Authors add a sentence in the Discussion that includes this potential caveat. I understand the reasoning the Authors use here - however, these are all assumptions, and one cannot exclude the possibility that BTZ also affected mammalian proteasome (or other pathway in mice) which has contributed to the effect observed in fungal cell size.

3. My new comment to the response to comment #18: I think the Authors should state in the text that the measured ROS and PRN refer to total amount (macrophages and fungal cells). 

Author Response

Dear review 1, 

Thanks for the comment. 

1. My new comment to the response to the initial comment #8: I suggest the Authors actually replace the "molecular signatures" with "transcriptional profile" - that way the reader will understand that here the Authors refer to transcriptional profile and not to some other (ill-defined) feature.

Thanks for the comment. We modified the text.

2. My new comment to the response to comment #12: I suggest the Authors add a sentence in the Discussion that includes this potential caveat. I understand the reasoning the Authors use here - however, these are all assumptions, and one cannot exclude the possibility that BTZ also affected mammalian proteasome (or other pathway in mice) which has contributed to the effect observed in fungal cell size.

Thanks for the comment. We modified the text in line 765-766.

3. My new comment to the response to comment #18: I think the Authors should state in the text that the measured ROS and PRN refer to total amount (macrophages and fungal cells).

Thanks for the comment. We modified the text in line 181-182.

Reviewer 2 Report

The authors have very respectfully and completely answered all my comments - kudos to them! I do not have any other concerns and I believe the manuscript is now clearer, stronger, and suitable for publication. 

Author Response

Dear review 2;

Thanks for the review. His comments were extremely important for the improvement of the manuscript.